# Learning to Compose Domain-Specific Transformations for Data Augmentation

**Alexander J. Ratner**,[*] **Henry R. Ehrenberg**,[*] **Zeshan Hussain**,
**Jared Dunnmon, Christopher Ré**
Stanford University
{ajratner,henryre,zeshanmh,jdunnmon,chrismre}@cs.stanford.edu

## Abstract

Data augmentation is a ubiquitous technique for increasing the size of labeled training sets by leveraging task-specific data transformations that preserve class labels. While it is often easy for domain experts to specify individual transformations, constructing and tuning the more sophisticated compositions typically needed to achieve state-of-the-art results is a time-consuming manual task in practice. We propose a method for automating this process by learning a generative sequence model over user-specified transformation functions using a generative adversarial approach. Our method can make use of arbitrary, non-deterministic transformation functions, is robust to misspecified user input, and is trained on unlabeled data. The learned transformation model can then be used to perform data augmentation for any end discriminative model. In our experiments, we show the efficacy of our approach on both image and text datasets, achieving improvements of 4.0 accuracy points on CIFAR-10, 1.4 F1 points on the ACE relation extraction task, and 3.4 accuracy points when using domain-specific transformation operations on a medical imaging dataset as compared to standard heuristic augmentation approaches.

## 1 Introduction

Modern machine learning models, such as deep neural networks, may have billions of free parameters and accordingly require massive labeled data sets for training. In most settings, labeled data is not available in sufficient quantities to avoid overfitting to the training set. The technique of artificially expanding labeled training sets by transforming data points in ways which preserve class labels – known as *data augmentation* – has quickly become a critical and effective tool for combatting this labeled data scarcity problem. Data augmentation can be seen as a form of *weak supervision*, providing a way for practitioners to leverage their knowledge of invariances in a task or domain. And indeed, data augmentation is cited as essential to nearly every state-of-the-art result in image classification [3, 7, 11, 24] (see Supplemental Materials), and is becoming increasingly common in other modalities as well [20].

Even on well studied benchmark tasks, however, the choice of data augmentation strategy is known to cause large variances in end performance and be difficult to select [11, 7], with papers often reporting their heuristically found parameter ranges [3]. In practice, it is often simple to formulate a large set of primitive transformation operations, but time-consuming and difficult to find the parameterizations and compositions of them needed for state-of-the-art results. In particular, many transformation operations will have vastly different effects based on parameterization, the set of other transformations they are applied with, and even their particular order of composition. For example, brightness and saturation enhancements might be destructive when applied together, but produce realistic images when paired with geometric transformations.

---

[*]Authors contributed equally

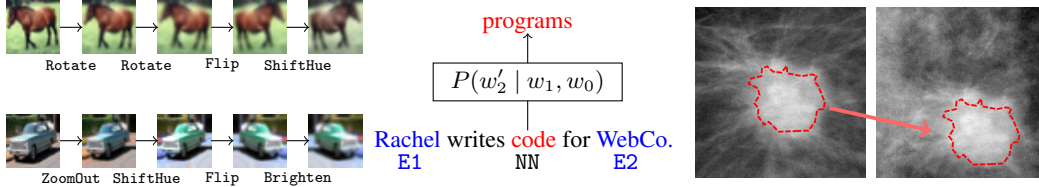

Figure 1: Three examples of transformation functions (TFs) in different domains: Two example sequences of incremental image TFs applied to CIFAR-10 images (*left*); a conditional word-swap TF using an externally trained language model and specifically targeting nouns (NN) between entity mentions (E1,E2) for a relation extraction task (*middle*); and an unsupervised segementation-based translation TF applied to mass-containing mammography images (*right*).

Given the difficulty of searching over this configuration space, the de facto norm in practice consists of applying one or more transformations in random order and with random parameterizations selected from hand-tuned ranges. Recent lines of work attempt to automate data augmentation entirely, but either rely on large quantities of labeled data [1, 21], restricted sets of simple transformations [8, 13], or consider only local perturbations that are not informed by domain knowledge [1, 22] (see Section 4). In contrast, our aim is to directly and flexibly leverage domain experts' knowledge of invariances as a valuable form of weak supervision in real-world settings where labeled training data is limited.

In this paper, we present a new method for data augmentation that directly leverages user domain knowledge in the form of transformation operations, and automates the difficult process of composing and parameterizing them. We formulate the problem as one of learning a generative sequence model over black-box *transformation functions (TFs)*: user-specified operators representing incremental transformations to data points that need not be differentiable nor deterministic. For example, TFs could rotate an image by a small degree, swap a word in a sentence, or translate a segmented structure in an image (Fig. 1). We then design a generative adversarial objective [9] which allows us to train the sequence model to produce transformed data points which are still within the data distribution of interest, using unlabeled data. Because the TFs can be stochastic or non-differentiable, we present a reinforcement learning-based training strategy for this model. The learned model can then be used to perform data augmentation on labeled training data for any end discriminative model.

Given the flexibility of our representation of the data augmentation process, we can apply our approach in many different domains, and on different modalities including both text and images. On a real-world mammography image task, we achieve a 3.4 accuracy point boost above randomly composed augmentation by learning to appropriately combine standard image TFs with domain-specific TFs derived in collaboration with radiology experts. Using novel language model-based TFs, we see a 1.4 F1 boost over heuristic augmentation on a text relation extraction task from the ACE corpus. And on a 10%-subsample of the CIFAR-10 dataset, we achieve a 4.0 accuracy point gain over a standard heuristic augmentation approach and are competitive with comparable semi-supervised approaches. Additionally, we show empirical results suggesting that the proposed approach is robust to misspecified TFs. Our hope is that the proposed method will be of practical value to practitioners and of interest to researchers, so we have open-sourced the code at `https://github.com/HazyResearch/tanda`.

## 2 Modeling Setup and Motivation

In the standard data augmentation setting, our aim is to expand a labeled training set by leveraging knowledge of class-preserving transformations. For a practitioner with domain expertise, providing individual transformations is straightforward. However, high performance augmentation techniques use *compositions* of finely tuned transformations to achieve state-of-the-art results [7, 3, 11], and heuristically searching over this space of all possible compositions and parameterizations for a new task is often infeasible. Our goal is to automate this task by learning to compose and parameterize a set of user-specified transformation operators in ways that are diverse but still preserve class labels.

In our method, transformations are modeled as sequences of incremental user-specified operations, called transformation functions (TFs) (Fig. 1). Rather than making the strong assumption that all the provided TFs preserve class labels, as existing approaches do, we assume a weaker form of class

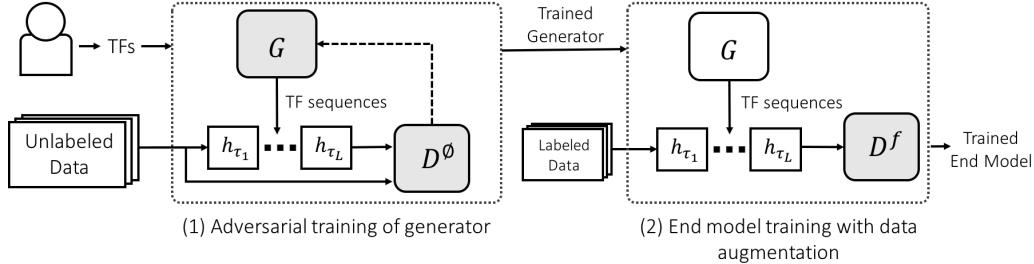

(1) Adversarial training of generator      (2) End model training with data augmentation

Figure 2: A high-level diagram of our method. Users input a set of transformation functions $h_1, ..., h_K$ and unlabeled data. A generative adversarial approach is then used to train a *null class* discriminator, $D^\emptyset$, and a generator, $G$, which produces TF sequences $h_{\tau_1}, ..., h_{\tau_L}$. Finally, the trained generator is used to perform data augmentation for an end discriminative model $D^f$.

invariance which enables us to use *unlabeled* data to learn a generative model over transformation sequences. We then propose two representative model classes to handle modeling both commutative and non-commutative transformations.

## 2.1 Augmentation as Sequence Modeling

In our approach, we represent transformations as sequences of incremental operations. In this setting, the user provides a set of $K$ TFs, $h_i : \mathcal{X} \mapsto \mathcal{X}$, $i \in [1, K]$. Each TF performs an incremental transformation: for example, $h_i$ could rotate an image by five degrees, swap a word in a sentence, or move a segmented tumor mass around a background mammography image (see Fig. 1). In order to accommodate a wide range of such user-defined TFs, we treat them as black-box functions which need not be deterministic nor differentiable.

This formulation gives us a tractable way to tune both the parameterization and composition of the TFs in a discretized but fine-grained manner. Our representation can be thought of as an implicit binning strategy for tuning parameterizations – e.g. a 15 degree rotation might be represented as three applications of a five-degree rotation TF. It also provides a direct way to represent compositions of multiple transformation operations. This is critical as a multitude of state-of-the-art results in the literature show the importance of using compositions of more than one transformations per image [7, 3, 11], which we also confirm experimentally in Section 5.

## 2.2 Weakening the Class-Invariance Assumption

Any data augmentation technique fundamentally relies on some assumption about the transformation operations' relation to the class labels. Previous approaches make the unrealistic assumption that all provided transformation operations preserve class labels for all data points. That is,

$$y(h_{\tau_L} \circ \ldots \circ h_{\tau_1}(x)) = y(x) \tag{1}$$

for label mapping function $y$, any sequence of TF indices $\tau_1, ..., \tau_L$, and *all* data points $x$.

This assumption puts a large burden of precise specification on the user, and based on our observations, is violated by many real-world data augmentation strategies. Instead, we consider a weaker modeling assumption. We assume that transformation operations will not map between classes, but might destructively map data points out of the distribution of interest entirely:

$$y(h_{\tau_L} \circ \ldots \circ h_{\tau_1}(x)) \in \{y(x), y_\emptyset\} \tag{2}$$

where $y_\emptyset$ represents an out-of-distribution *null class*. Intuitively, this weaker assumption is motivated by the categorical image classification setting, where we observe that transformation operations provided by the user will almost never turn, for example, a plane into a car, but may often turn a plane into an indistinguishable "garbage" image (Fig. 3). We are the first to consider this weaker invariance assumption, which we believe more closely matches various practical data augmentation settings of interest. In Section 5, we also provide empirical evidence that this weaker assumption is useful in binary classification settings and over modalities other than image data. Critically, it also enables us to learn a model of TF sequences using unlabeled data alone.

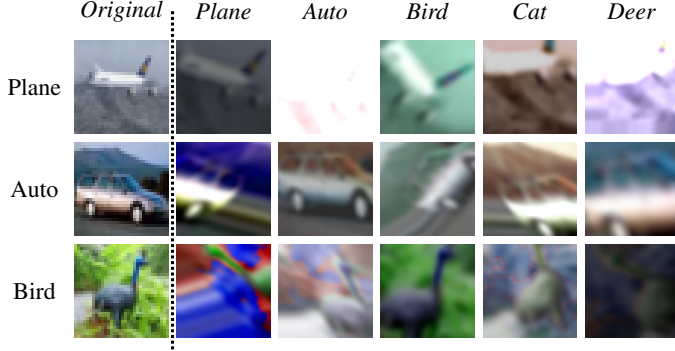
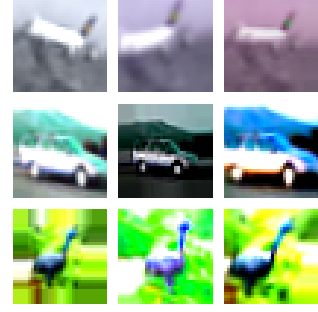

Figure 3: Our modeling assumption is that transformations may map out of the natural distribution of interest, but will rarely map *between* classes. As a demonstration, we take images from CIFAR-10 (each row) and randomly search for a transformation sequence that best maps them to a different class (each column), according to a trained discriminative model. The matches rarely resemble the target class but often no longer look like "normal" images at all. Note that we consider a fixed set of user-provided TFs, not adversarially selected ones.

Figure 4: Some example transformed images generated using an augmentation generative model trained using our approach. Note that this is not meant as a comparison to Fig. 3.

## 2.3 Minimizing Null Class Mappings Using Unlabeled Data

Given assumption (2), our objective is to learn a model $G_\theta$ which generates sequences of TF indices $\tau \in \{1, K\}^L$ with fixed length $L$, such that the resulting TF sequences $h_{\tau_1}, ..., h_{\tau_L}$ are not likely to map data points into $y_\emptyset$. Crucially, this does not involve using the class labels of any data points, and so we can use unlabeled data. Our goal is then to minimize the the probability of a generated sequence mapping unlabeled data points into the null class, with respect to $\theta$:

$$J_\emptyset = \mathbb{E}_{\tau \sim G_\theta} \mathbb{E}_{x \sim \mathcal{U}} \left[ P(y(h_{\tau_L} \circ \ldots \circ h_{\tau_1}(x)) = y_\emptyset) \right] \tag{3}$$

where $\mathcal{U}$ is some distribution of unlabeled data.

**Generative Adversarial Objective**  In order to approximate $P(y(h_{\tau_1} \circ \ldots \circ h_{\tau_L}(x)) = y_\emptyset)$, we jointly train the generator $G_\theta$ and a discriminative model $D_\phi^\emptyset$ using a generative adversarial network (GAN) objective [9], now minimizing with respect to $\theta$ and maximizing with respect to $\phi$:

$$\tilde{J}_\emptyset = \mathbb{E}_{\tau \sim G_\theta} \mathbb{E}_{x \sim \mathcal{U}} \left[ \log(1 - D_\phi^\emptyset(h_{\tau_L} \circ \ldots \circ h_{\tau_1}(x))) \right] + \mathbb{E}_{x' \sim \mathcal{U}} \left[ \log(D_\phi^\emptyset(x')) \right] \tag{4}$$

As in the standard GAN setup, the training procedure can be viewed as a minimax game in which the discriminator's goal is to assign low values to transformed, out-of-distribution data points and high values to real in-distribution data points, while simultaneously, the generator's goal is to generate transformation sequences which produce data points that are indistinguishable from real data points according to the discriminator. For $D_\phi^\emptyset$, we use an all-convolution CNN as in [23]. For further details, see Supplemental Materials.

**Diversity Objective**  An additional concern is that the model will learn a variety of null transformation sequences (e.g. rotating first left than right repeatedly). Given the potentially large state-space of actions, and the black-box nature of the user-specified TFs, it seems infeasible to hard-code sets of inverse operations to avoid. To mitigate this, we instead consider a second objective term:

$$J_d = \mathbb{E}_{\tau \sim G_\theta} \mathbb{E}_{x \sim \mathcal{U}} \left[ d(h_{\tau_L} \circ \ldots \circ h_{\tau_1}(x), x) \right] \tag{5}$$

where $d : \mathcal{X} \times \mathcal{X} \to \mathbb{R}$ is some distance function. For $d$, we evaluated using both distance in the raw input space, and in the feature space learned by the final pre-softmax layer of the discriminator $D_\phi^\emptyset$. Combining eqns. 4 and 5, our final objective is then $J = \tilde{J}_\emptyset + \alpha J_d^{-1}$ where $\alpha > 0$ is a hyperparameter. We minimize $J$ with respect to $\theta$ and maximize with respect to $\phi$.

## 2.4 Modeling Transformation Sequences

We now consider two model classes for $G_\theta$:

**Independent Model**   We first consider a *mean field* model in which each sequential TF is chosen independently. This reduces our task to one of learning $K$ parameters, which we can think of as representing the task-specific "accuracies" or "frequencies" of each TF. For example, we might want to learn that elastic deformations or swirls should only rarely be applied to images in CIFAR-10, but that small rotations can be applied frequently. In particular, a mean field model also provides a simple way of effectively learning stochastic, discretized parameterizations of the TFs. For example, if we have a TF representing five-degree rotations, `Rotate5Deg`, a marginal value of $P_{G_\theta}(\texttt{Rotate5Deg}) = 0.1$ could be thought of as roughly equivalent to learning to rotate $0.5L$ degrees on average.

**State-Based Model**   There are important cases, however, where the independent representation learned by the mean field model could be overly limited. In many settings, certain TFs may have very different effects depending on which other TFs are applied with them. As an example, certain similar pairs of image transformations might be overly lossy when applied together, such as a blur and a zoom operation, or a brighten and a saturate operation. A mean field model could not represent such disjunctions as these. Another scenario where an independent model fails is where the TFs are non-commutative, such as with lossy operators (e.g. image transformations which use aliasing). In both of these cases, modeling the sequences of transformations could be important. Therefore we consider a long short-term memory (LSTM) network as as a representative sequence model. The output from each cell of the network is a distribution over the TFs. The next TF in the sequence is then sampled from this distribution, and is fed as a one-hot vector to the next cell in the network.

## 3 Learning a Transformation Sequence Model

The core challenge that we now face in learning $G_\theta$ is that it generates sequences over TFs which are not necessarily differentiable or deterministic. This constraint is a critical facet of our approach from the usability perspective, as it allows users to easily write TFs as black-box scripts in the language of their choosing, leveraging arbitrary subfunctions, libraries, and methods. In order to work around this constraint, we now describe our model in the syntax of reinforcement learning (RL), which provides a convenient framework and set of approaches for handling computation graphs with non-differentiable or stochastic nodes [27].

**Reinforcement Learning Formulation**   Let $\tau_i$ be the index of the $i$th TF applied, and $\tilde{x}_i$ be the resulting incrementally transformed data point. Then we consider $s_t = (x, \tilde{x}_1, \tilde{x}_2, ..., \tilde{x}_t, \tau_1, ...., \tau_t)$ as the state after having applied $t$ of the incremental TFs. Note that we include the incrementally transformed data points $\tilde{x}_1, ..., \tilde{x}_t$ in $s_t$ since the TFs may be stochastic. Each of the model classes considered for $G_\theta$ then uses a different *state representation $\hat{s}$*. For the mean field model, the state representation used is $\hat{s}_t^{\text{MF}} = \emptyset$. For the LSTM model, we use $\hat{s}_t^{\text{LSTM}} = \text{LSTM}(\tau_t, s_{t-1})$, the state update operation performed by a standard LSTM cell parameterized by $\theta$.

**Policy Gradient with Incremental Rewards**   Let $\ell_t(x, \tau) = \log(1 - D_\phi^\emptyset(\tilde{x}_t))$ be the *cumulative loss* for a data point $x$ at step $t$, with $\ell_0(x) = \ell_0(x, \tau) \equiv \log(1 - D_\phi^\emptyset(x))$. Let $R(s_t) = \ell_t(x, \tau) - \ell_{t-1}(x, \tau)$ be the *incremental reward*, representing the difference in discriminator loss at incremental transformation step $t$. We can now recast the first term of our objective $\tilde{J}_\emptyset$ as an expected sum of incremental rewards:

$$U(\theta) \equiv \mathbb{E}_{\tau \sim G_\theta} \mathbb{E}_{x \sim \mathcal{U}} \left[ \log(1 - D_\phi^\emptyset(h_{\tau_1} \circ \ldots \circ h_{\tau_L}(x))) \right] = \mathbb{E}_{\tau \sim G_\theta} \mathbb{E}_{x \sim \mathcal{U}} \left[ \ell_0(x) + \sum_{t=1}^{L} R(s_t) \right] \tag{6}$$

We omit $\ell_0$ in practice, equivalent to using the loss of $x$ as a baseline term. Next, let $\pi_\theta$ be the stochastic transition policy implicitly defined by $G_\theta$. We compute the recurrent policy gradient [32]

of the objective $U(\theta)$ as:

$$\nabla_\theta U(\theta) = \mathbb{E}_{\tau \sim G_\theta} \mathbb{E}_{x \sim \mathcal{U}} \left[ \sum_{t=1}^{L} R(s_t) \nabla_\theta \log \pi_\theta(\tau_t \mid \hat{s}_{t-1}) \right] \qquad (7)$$

Following standard practice, we approximate this quantity by sampling batches of $n$ data points and $m$ sampled action sequences per data point. We also use standard techniques of discounting with factor $\gamma \in [0, 1]$ and considering only future rewards [12]. See Supplemental Materials for details.

## 4 Related Work

We now review related work, both to motivate comparisons in the experiments section and to present complementary lines of work.

**Heuristic Data Augmentation**   Most state-of-the-art image classification pipelines use some limited form of data augmentation [11, 7]. This generally consists of applying crops, flips, or small affine transformations, in fixed order or at random, and with parameters drawn randomly from hand-tuned ranges. In addition, various studies have applied heuristic data augmentation techniques to modalities such as audio [31] and text [20]. As reported in the literature, the selection of these augmentation strategies can have large performance impacts, and thus can require extensive selection and tuning by hand [3, 7] (see Supplemental Materials as well).

**Interpolation-Based Techniques**   Some techniques have explored generating augmented training sets by interpolating between labeled data points. For example, the well-known SMOTE algorithm applies this basic technique for oversampling in class-imbalanced settings [2], and recent work explores using a similar interpolation approach in a learned feature space [5]. [13] proposes learning a class-conditional model of diffeomorphisms interpolating between nearest-neighbor labeled data points as a way to perform augmentation. We view these approaches as complementary but orthogonal, as our goal is to directly exploit user domain knowledge of class-invariant transformation operations.

**Adversarial Data Augmentation**   Several lines of recent work have explored techniques which can be viewed as forms of data augmentation that are adversarial with respect to the end classification model. In one set of approaches, transformation operations are selected adaptively from a given set in order to maximize the loss of the end classification model being trained [30, 8]. These procedures make the strong assumption that all of the provided transformations will preserve class labels, or use bespoke models over restricted sets of operations [28]. Another line of recent work has showed that augmentation via small adversarial linear perturbations can act as a regularizer [10, 22]. While complimentary, this work does not consider taking advantage of non-local transformations derived from user knowledge of task or domain invariances.

Finally, generative adversarial networks (GANs) [9] have recently made great progress in learning complete data generation models from unlabeled data. These can be used to augment labeled training sets as well. Class-conditional GANs [1, 21] generate artificial data points but require large sets of labeled training data to learn from. Standard unsupervised GANs can be used to generate additional out-of-class data points that can then augment labeled training sets [25, 29]. We compare our proposed approach with these methods empirically in Section 5.

## 5 Experiments

We experimentally validate the proposed framework by learning augmentation models for several benchmark and real-world data sets, exploring both image recognition and natural language understanding tasks. Our focus is on the performance of end classification models trained on labeled datasets augmented with our approach and others used in practice. We also examine robustness to user misspecification of TFs, and sensitivity to core hyperparameters.

### 5.1 Datasets and Transformation Functions

**Benchmark Image Datasets**   We ran experiments on the MNIST [18] and CIFAR-10 [17] datasets, using only a subset of the class labels to train the end classification models and treating the rest

as unlabeled data. We used a generic set of TFs for both MNIST and CIFAR-10: small rotations, shears, central swirls, and elastic deformations. We also used morphologic operations for MNIST, and adjustments to hue, saturation, contrast, and brightness for CIFAR-10.

**Benchmark Text Dataset**   We applied our approach to the *Employment* relation extraction subtask from the NIST Automatic Content Extraction (ACE) corpus [6], where the goal is to identify mentions of employer-employee relations in news articles. Given the standard class imbalance in information extraction tasks like this, we used data augmentation to oversample the minority positive class. The flexibility of our TF representation allowed us to take a straightforward but novel approach to data augmentation in this setting. We constructed a trigram language model using the ACE corpus and Reuters Corpus Volume I [19] from which we can sample a word conditioned on the preceding words. We then used this model as the basis for a set of TFs that select words to swap based on the part-of-speech tag and location relative to entities of interest (see Supplemental Materials for details).

**Mammography Tumor-Classification Dataset**   To demonstrate the effectiveness of our approach on real-world applications, we also considered the task of classifying benign versus malignant tumors from images in the Digital Database for Screening Mammography (DDSM) dataset [15, 4, 26], which is a class-balanced dataset consisting of 1506 labeled mammograms. In collaboration with domain experts in radiology, we constructed two basic TF sets. The first set consisted of standard image transformation operations subselected so as not to break class-invariance in the mammography setting. For example, brightness operations were excluded for this reason. The second set consisted of both the first set as well as several novel segmentation-based transplantation TFs. Each of these TFs utilized the output of an unsupervised segmentation algorithm to isolate the tumor mass, perform a transformation operation such as rotation or shifting, and then stitch it into a randomly-sampled benign tissue image. See Fig. 1 (right panel) for an illustrative example, and Supplemental Materials for further details.

## 5.2   End Classifier Performance

We evaluated our approach by using it to augment labeled training sets for the tasks mentioned above, and show that we achieve strong gains over heuristic baselines. In particular, for a given set of TFs, we evaluate the performance of mean field (*MF*) and LSTM generators trained using our approach against two standard data augmentation techniques used in practice. The first (*Basic*) consists of applying random crops to images, or performing simple minority class duplication for the ACE relation extraction task. The second (*Heur.*) is the standard heuristic approach of applying random compositions of the given set of transformation operations, the most common technique used in practice [3, 11, 14]. For both our approaches (*MF* and *LSTM*) and *Heur.*, we additionally use the same random cropping technique as in the *Basic* approach. We present these results in Table 1, where we report test set accuracy (or F1 score for ACE), and use a random subsample of the available labeled training data. Additionally, we include an extra row for the DDSM task highlighting the impact of adding domain-specific (*DS*) TFs – the segmentation-based operations described above – on performance.

In Table 2 we additionally compare to two related generative-adversarial methods, the Categorical GAN (CatGAN) [29], and the semi-supervised GAN (SS-GAN) from [25]. Both of these methods use GAN-based architectures trained on unlabeled data to generate new out-of-class data points with which to augment a labeled training set. Following their protocol for CIFAR-10, we train our generator on the full set of unlabeled data, and our end discriminator on ten disjoint random folds of the labeled training set not including the validation set (i.e. $n = 4000$ each), averaging the results.

In all settings, we train our TF sequence generator on the full set of unlabeled data. We select a fixed sequence length for each task via an initial calibration experiment (Fig. 5b). We use $L = 5$ for ACE, $L = 7$ for DDSM + DS, and $L = 10$ for all other tasks. We note that our findings here mirrored those in the literature, namely that compositions of multiple TFs lead to higher end model accuracies. We selected hyperparameters of the generator via performance on a validation set. We then used the trained generator to transform the entire training set at each epoch of end classification model training. For MNIST and DDSM we use a four-layer all-convolutional CNN, for CIFAR10 we use a 56-layer ResNet [14], and for ACE we use a bi-directional LSTM. Additionally, we incorporate a basic transformation regularization term as in [24] (see Supplemental Materials), and train for the last ten epochs without applying any transformations as in [11]. In all cases, we use hyperparameters as

| Task | % | None | Basic | Heur. | MF | LSTM |
|------|---|------|-------|-------|-----|------|
| MNIST | 1 | 90.2 | 95.3 | 95.9 | 96.5 | **96.7** |
|  | 10 | 97.3 | 98.7 | 99.0 | **99.2** | 99.1 |
| CIFAR-10 | 10 | 66.0 | 73.1 | 77.5 | 79.8 | **81.5** |
|  | 100 | 87.8 | 91.9 | 92.3 | **94.4** | 94.0 |
| ACE (F1) | 100 | 62.7 | 59.9 | 62.8 | 62.9 | **64.2** |
| DDSM | 10 | 57.6 | 58.8 | 59.3 | 58.2 | 61.0 |
| DDSM + DS |  |  |  | 53.7 | 59.9 | **62.7** |

Table 1: Test set performance of end models trained on subsamples of the labeled training data (%), not including validation splits, using various data augmentation approaches. *None* indicates performance with no augmentation. All tasks are measured in accuracy, except ACE which is measured by F1 score.

| Model | Acc. (%) |
|-------|----------|
| CatGAN | $80.42 \pm 0.58$ |
| SS-GAN | $81.37 \pm 2.32$ |
| **LSTM** | $81.47 \pm 0.46$ |

Table 2: Reported end model accuracies, averaged across 10% subsample folds, on CIFAR-10 for comparable GAN methods.

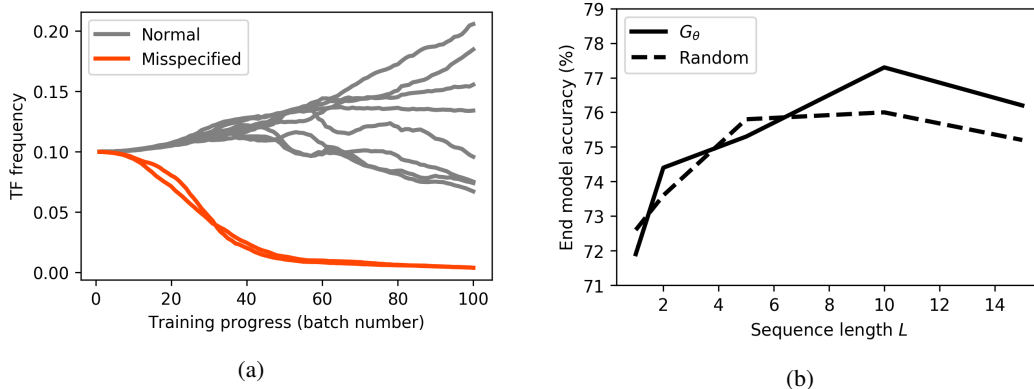

(a)

(b)

Figure 5: (a) Learned TF frequency parameters for misspecified and normal TFs on MNIST. The mean field model correctly learns to avoid the misspecified TFs. (b) Larger sequence lengths lead to higher end model accuracy on CIFAR-10, while random performs best with shorter sequences, according to a sequence length calibration experiment.

reported in the literature. For further details of generator and end model training see the Supplemental Materials.

We see that across the applications studied, our approach outperforms the heuristic data augmentation approach most commonly used in practice. Furthermore, the LSTM generator outperforms the simple mean field one in most settings, indicating the value of modeling sequential structure in data augmentation. In particular, we realize significant gains over standard heuristic data augmentation on CIFAR-10, where we are competitive with comparable semi-supervised GAN approaches, but with significantly smaller variance. We also train the same CIFAR-10 end model using the full labeled training dataset, and again see strong relative gains (2.1 pts. in accuracy over heuristic), coming within 2.1 points of the current state-of-the-art [16] using our much simpler end model.

On the ACE and DDSM tasks, we also achieve strong performance gains, showing the ability of our method to productively incorporate more complex transformation operations from domain expert users. In particular, in DDSM we observe that the addition of the segmentation-based TFs causes the heuristic augmentation approach to perform significantly worse, due to a large number of new failure modes resulting from combinations of the segmentation-based TFs – which use gradient-based blending – and the standard TFs such as zoom and rotate. In contrast, our LSTM model learns to avoid these destructive subsequences and achieves the highest score, resulting in a 9.0 point boost over the comparable heuristic approach.

**Robustness to TF Misspecification** One of the high-level goals of our approach is to enable an easier interface for users by not requiring that the TFs they specify be completely class-preserving. The lack of any assumption of well-specified transformation operations in our approach, and the strong empirical performance realized, is evidence of this robustness. To additionally illustrate the robustness of our approach to misspecified TFs, we train a mean field generator on MNIST using the standard TF set, but with two TFs (shear operations) parameterized so as to map almost all images to the null class. We see in Fig. 5a that the generator learns to avoid applying the misspecified TFs (red lines) almost entirely.

## 6 Conclusion and Future Work

We presented a method for learning how to parameterize and compose user-provided black-box transformation operations used for data augmentation. Our approach is able to model arbitrary TFs, allowing practitioners to leverage domain knowledge in a flexible and simple manner. By training a generative sequence model over the specified transformation functions using reinforcement learning in a GAN-like framework, we are able to generate realistic transformed data points which are useful for data augmentation. We demonstrated that our method yields strong gains over standard heuristic approaches to data augmentation for a range of applications, modalities, and complex domain-specific transformation functions. There are many possible future directions of research for learning data augmentation strategies in the proposed model, such as conditioning the generator's stochastic policy on a featurized version of the data point being transformed, and generating TF sequences of dynamic length. More broadly, we are excited about further formalizing data augmentation as a novel form of weak supervision, allowing users to directly encode domain knowledge about invariants into machine learning models.

**Acknowledgements** We would like to thank Daniel Selsam, Ioannis Mitliagkas, Christopher De Sa, William Hamilton, and Daniel Rubin for valuable feedback and conversations. We gratefully acknowledge the support of the Defense Advanced Research Projects Agency (DARPA) SIMPLEX program under No. N66001-15-C-4043, the DARPA D3M program under No. FA8750-17-2-0095, DARPA programs No. FA8750-12-2-0335 and FA8750-13-2-0039, DOE 108845, National Institute of Health (NIH) U54EB020405, the Office of Naval Research (ONR) under awards No. N000141210041 and No. N000141310129, the Moore Foundation, the Okawa Research Grant, American Family Insurance, Accenture, Toshiba, and Intel. This research was also supported in part by affiliate members and other supporters of the Stanford DAWN project: Intel, Microsoft, Teradata, and VMware. This material is based on research sponsored by DARPA under agreement number FA8750-17-2-0095. The U.S. Government is authorized to reproduce and distribute reprints for Governmental purposes notwithstanding any copyright notation thereon. Any opinions, findings, and conclusions or recommendations expressed in this material are those of the authors and do not necessarily reflect the views, policies, or endorsements, either expressed or implied, of DARPA, AFRL, NSF, NIH, ONR, or the U.S. Government.

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
