[Supplementary Material · tanda_camera_ready_with_appendix.pdf]

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

# A   Additional Background

## A.1   Review of Data Augmentation Use in State-of-the-Art

To underscore both the omnipresence and diversity of heuristic data augmentation in practice, we compiled a list of the top ten models for the well documented CIFAR-10 and CIFAR-100 tasks (Table 3). We see that in 10 out of 10 of the top CIFAR-10 results and 9 out of 10 of the top CIFAR-100 results use data augmentation, for average boosts (when reported) of 3.71 and 13.39 points in accuracy, respectively. Moreover, we see that while some sets of papers inherit a simple data augmentation strategy from prior work (in particular, all the recent ResNet variants), there are still a large variety of approaches. And in general, the particular choice of data augmentation strategy is widely reported to have large effects on performance.

Note that the below table is compiled from a well-known online compendium [2] and from the latest CVPR best paper [17] (indicated by a *) which achieves new state-of-the-art results. We compile it for illustrative purposes and it is not necessarily comprehensive. Also note that we selected CIFAR-10/100 both as a representative and well-studied task, but also due to the availability of published results. For competitions such as ImageNet, although data augmentation is widely reported to be critical, many top results are reported very opaquely, with little described about implementation details such as data augmentation.

# B   Additional Experiments

## B.1   Synthetic Data Examples

**Simple Synthetic Setup**   As both a diagnostic tool and as a simple way to probe the properties of our approach, we construct a simple synthetic dataset consisting of points in two dimensions, uniformly selected in a ball of radius $r = 1$ around the origin. We then consider various displacement vectors as our TFs. We consider the same two generative models as in our main experiments – mean

---
[2]

| Dataset | Pos. | Name | Err. w/DA | Err. w/o DA | Notes |
|---|---|---|---|---|---|
| CIFAR-10 | 1 | DenseNet | 3.46 | - | Random shifts, flips |
| | 2 | Fractional Max-Pooling | 3.47 | - | Randomized mix of translations, rotations, reflections, stretching, shearing, and random RGB color shift operations |
| | 3* | Wide ResNet | 4.17 | - | Random shifts, flips |
| | 4 | Striving for Simplicity: The All Convolutional Net | 4.41 | 9.08 | "Heavy" augmentation: images expanded, then scaled, rotated, color shifted randomly |
| | 5* | FractalNet | 4.60 | 7.33 | Random shifts, flips |
| | 6* | ResNet (1001-Layer) | 4.62 | 10.56 | Random shifts, flips |
| | 7* | ResNet with Stochastic Depth (1202-Layer) | 4.91 | - | Random shifts, flips |
| | 8 | All You Need is a Good Init | 5.84 | - | Random shifts, flips |
| | 9 | Generalizing Pooling Functions in Convolutional Neural Networks: Mixed, Gated, and Tree | 6.05 | 7.62 | Flips, random shifts, other simple ones |
| | 10 | Spatially-Sparse Convolutional Neural Networks | 6.28 | - | Affine transformations |
| CIFAR-100 | 1* | DenseNet | 17.18 | - | Random shifts, flips |
| | 2* | Wide ResNets | 20.50 | - | Random shifts, flips |
| | 3* | ResNet (1001-Layer) | 22.71 | 33.47 | Random shifts, flips |
| | 4* | FractalNet | 23.30 | 35.34 | Random shifts, flips |
| | 5 | Fast and Accurate Deep Network Learning by Exponential Linear Units | - | 24.28 | |
| | 6 | Spatially-Sparse Convolutional Neural Networks | 24.3 | - | Affine transformations |
| | 7* | ResNet with Stochastic Depth (1202-Layer) | 24.58 | 37.80 | Random shifts, flips |
| | 8 | Fractional Max-Pooling | 26.39 | - | Randomized mix of translations, rotations, reflections, stretching, and shearing operations, and random RGB color shifts |
| | 9* | ResNet (110-Layer) | 27.22 | 44.74 | Random shifts, flips |
| | 10 | Scalable Bayesian Optimization Using Deep Neural Networks | 27.4 | - | Hue, saturation, scalings, horizontal flips |

Table 3: Current state-of-the-art image classification models as ranked by reported performance on the CIFAR-10 and CIFAR-100 tasks, and their error with (*Err. w/ DA*) and without (*Err. w/o DA*) data augmentation. We include both scores and particular data augmentation techniques when reported, although the latter is rarely reported with great precision.

field and LSTM – and use either a basic fully-connected two-layer neural network or an oracle discriminator $f(x) = 1\{||x|| < 1\}$.

(a) Mean field model on TF set 1     (b) Mean field model on TF set 2     (c) LSTM model on TF set 2

Figure 6: Original data points (blue) are transformed using sequences of vector displacement TFs ($L = 10$) drawn from $G_\theta$, producing augmented data points (red). $G_\theta$ is either a mean field model or an LSTM, trained with an orcale discriminator $D^\emptyset$ for 15 epochs.

**Synthetic Experiments** In this setting, we define $y_\emptyset(x) = 1\{||x|| \geq 1\}$, and consider two different TF sets:

1. *Good vs. Bad TFs:* In a first toy scenario we consider TFs which are vector displacements of random direction, with magnitude drawn from one of two distributions, $\mathcal{N}(\mu_1, \sigma_1)$ or $\mathcal{N}(\mu_2, \sigma_2)$, where $\mu_1 > 1 > \mu_2$. In other words, the model should learn not to select certain individual TFs.
2. *Lossy TFs:* We consider a second toy setting where random-direction displacement TFs have magnitude drawn uniformly from $\mathcal{N}(\mu, \sigma)$, $\mu < 1$; however the magnitude of each TF decays exponentially with the distance a point is outside of the unit ball. This simulates the setting where TFs are irrecoverably lossy when applied in certain sequences.

As expected, we see that while the mean field model is able to model the first setting (Figure 6a), it fails to adequately represent the second one (Figure 6b), whereas the RNN model is able to (Figure 6c).

## B.2 Robustness to Transformed Test Data

We run a simple experiment to test the robustness of the trained end classification models to the individual TFs in the TF sets used. Specifically, on CIFAR-10 we create ten transformed copies a 10% subsample of the test data by transforming with a single TF, and then test the end model on this set. We compare our approach with heuristic random augmentation and no data augmentation of the model during training, and consider rotations, zooms, shears, and hue shifts. Results are presented in Figure 7.

Figure 7: Accuracy scores on random 10% subsamples of test data (dotted lines) and on versions augmented with a single transformation (vertical bars) with parameters drawn uniformly at random.

We can consider evaluating the results in terms of absolute robustness – i.e. model accuracy – and relative robustness, i.e. the change in model score when applied to the transformed test set. Roughly we see that our approach is most absolutely robust. Random appears to be most relatively robust, in particular on larger transformations, which we hypothesize our approach mostly learned to avoid applying during training.

## C  Reinforcement Learning Formulation Details

### C.1  Variance reduction methods

In Section 3, the vanilla policy gradient of our objective was given as

$$\nabla_\theta U(\theta) = \mathbb{E}_{\tau \sim G_\theta} \mathbb{E}_{x \sim \mathcal{U}} \left[ \sum_{t=1}^{L} R(s_t) \nabla_\theta \log \pi_\theta(\tau_t \mid \hat{s}_{t-1}) \right]$$

Noting that actions $\tau_t$ only impact future outcomes, following standard practice, we apply only future rewards in order to reduce variance:

$$\nabla_\theta U(\theta) = \mathbb{E}_{\tau \sim G_\theta} \mathbb{E}_{x \sim \mathcal{U}} \left[ \sum_{t=1}^{L} \nabla_\theta \log \pi_\theta(\tau_t \mid \hat{s}_{t-1}) \sum_{t'=t}^{L} \gamma^{t'-t} R(s_{t'}) \right]$$

where $\gamma \in [0, 1]$ is a discounting factor. Additionally, we use a baseline term $b_t$ to reduce variance when estimating $\nabla_\theta U(\theta)$:

$$\nabla_\theta U(\theta) = \mathbb{E}_{\tau \sim G_\theta} \mathbb{E}_{x \sim \mathcal{U}} \left[ \sum_{t=1}^{L} \nabla_\theta \log \pi_\theta(\tau_t \mid \hat{s}_{t-1}) \left( \left( \sum_{t'=t}^{L} \gamma^{t'-t} R(\hat{s}_{t'}) \right) - b_t \right) \right]$$

### C.2  Policy gradient estimation

Using a batch of $n$ data points and $m$ sampled action sequences per data point – given by the state representations $\{\hat{s}^{(i,j)}\}$ and action sequences $\{\tau^{(i,j)}\}$ – the gradient estimate is computed as:

$$\nabla_\theta \hat{U}(\theta) = \frac{1}{nm} \sum_{i=1}^{n} \sum_{j=1}^{m} \left[ \sum_{t=1}^{L} \nabla_\theta \log \pi_\theta(\tau_t^{(i,j)} \mid \hat{s}_{t-1}^{(i,j)}) \left( \left( \sum_{t'=t}^{L} \gamma^{t'-t} R(\hat{s}_{t'}^{(i,j)}) \right) - \hat{b}_t \right) \right]$$

where the baseline term $\hat{b}_t$ is also computed using the batch:

$$\hat{b}_t = \frac{1}{nm} \sum_{i=1}^{n} \sum_{j=1}^{m} \sum_{t'=t}^{L} \gamma^{t'-t} R\left(\hat{s}_{t'}^{(i,j)}\right)$$

In our experiments, we fixed $n = 32$ and $m = 5$.

## D  Experimental Details

### D.1  Benchmark Image Datasets

We use the MNIST dataset with 5000 training data points used as a validation set. We use the following TFs:

- Rotation (2.5, -2.5, 5, -5, 10, and -10 degrees)
- Zoom (0.9x, 1.1x)
- Shear (0.1, -0.1, 0.2, -0.2, 0.4, and -0.4 degrees)
- Swirl (0.1, -0.1, 0.2, -0.2, 0.4, and -0.4 degrees)
- Random elastic deformations ($\alpha = 1.0$, 1.25, and 1.5)
- Erosion

- Dilation

For the CIFAR-10 dataset, we use the following TFs:

- Rotation (2.5, -2.5, 5, -5 degrees)
- Zoom (0.9x, 1.1x, 0.75x, 1.25x)
- Shear (0.1, -0.1, 0.25, and -0.25 degrees)
- Swirl (0.1, -0.1, 0.25, -0.25 degrees)
- Hue Shift (by 0.1, -0.1, 0.25, and -0.25)
- Enhance contrast (by 0.75, 1.25, 0.5, and 1.5)
- Enhance brightness (by 0.75, 1.25, 0.5, and 1.5)
- Enhance color (by 0.75, 1.25, 0.5, and 1.5)
- Horizontal flip

For both datasets, we also applied random padding (by 4 pixels on each side) followed by random crops back to the original dimensions during training. We note that the choice of certain TFs to use in certain datasets was deliberate – for example, we would not expect horizontal flips or hue shifts to be appropriate in MNIST, or erosion and dilation to be useful in CIFAR-10. However, the particular choice of parameterizations was mainly due to disjoint implementations of the two experiments. For further details of the TF implementations used, see our code, which will be open-sourced after the review process.

## D.2 Benchmark Text Dataset

The ACE corpus consists of news articles and broadcast transcripts, all of which are pretagged with entity mentions. The objective of the Employment relation subtask is to extract *Person-Organization* entity pairs which are implied to have an affiliation in the text. We pose this as a binary classification problem by first identifying relation *candidates*: any pair of *Person-Organization* entities which occur in the same sentence. As noted in Section 5, there are far more true negative candidates than true positive candidates. The end model is trained to classify relation candidates as either true or false relations based on the raw text of the sentence in which they occur.

The language model described in Section 5 was constructed by recording counts of unigrams following each unique trigram, bigram, and unigram in the corpus. Laplace smoothing was applied to the counts, and basic filtering was applied to the $n$-grams. The sampler falls back to using bigrams then unigrams if the trigram preceding the word we want to swap was filtered out of the corpus. We used the following TFs in all experiments:

- Replace a noun to the left of both entities
- Replace a noun between the two entities
- Replace a noun to the right of both entities
- Replace a verb to the left of both entities
- Replace a verb between the two entities
- Replace a verb to the right of both entities
- Replace an adjective to the left of both entities
- Replace an adjective between the two entities
- Replace an adjective to the right of both entities

## D.3 DDSM Mammography Task

We use the following transformation functions:

1. Rotate Image: Rotate the entire mammogram by a deterministic angle $\theta$. Tumor geometry is fundamentally invariant to 2-D orientation. TF run with $\theta \in [-5^o, -2.5^o, 2.5^o, 5^o]$.

2. Zoom Image: Zoom in on the entire mammogram by a deterministic factor $\gamma$. Tumor classification is insensitive to $\gamma$ for $\gamma$ close to one. TF run with $\gamma \in \{0.98, 1.02\}$

3. Enhance Image Contrast: Enhance contrast values of grayscale image by a deterministic factor $\gamma$. Tumor classification is insensitive to $\gamma$ for $\gamma$ close to one. TF run with $\gamma \in \{0.95, 1.05\}$

4. Translate and Transplant Image: Extract pixels within mass segmentation. Perform translation of a bounding box of side length $N$ pixels about mass center by a deterministic vector $\hat{g}$. Transplant the translated bounding box containing the mass onto a randomly sampled normal tissue image using Poisson blending. Retains information about the mass itself within the context of a different set of normal tissue background. The bounding box also retains information about the tissue in the tumor near field. TF run with $N = 10$, $\hat{g} \in \{(-3, 0), (3, 0), (0, -3), (0, 3), (0, 0)\}$.

5. Rotate and Transplant Image: Extract pixels within mass segmentation. Perform rotation of a bounding box of side length $N$ pixels about mass center by a deterministic angle $\theta$. Transplant the rotated bounding box containing the mass onto a randomly sampled normal tissue image using Poisson blending [24]. Retains information about the mass itself within the context of a different set of normal tissue background. The bounding box also retains information about the tissue in the tumor near field. TF run with $N = 10$, $\theta \in \{-5^o, -2.5^o, 2.5^o, 5^o\}$.

Note that for the Poisson blending TFs, it is important that the translation and rotation domains be specified such that excessive proximity to the boundary of the destination image does not introduce spurious gradient information into the blended image.

## D.4   Details of Generative Adversarial Network Models

All models, both for the generator training as described in this section, and the end classification model training described next, were implemented in Tensorflow [3].

**Discriminator**   For image tasks, the discriminator used in the training of the generator in our approach was the same model as in [25], an all-convolutional CNN with four convolutional layers and leaky ReLU activations. For the text task, we used a unidirectional RNN with basic LSTM cells.

**Mean Field Model**   The mean field model is represented simply as a length $K$ vector of unbounded variables, where $K$ is the number of TFs. Applying the softmax function to this vector yields the TF sampling distribution.

**LSTM**   In the LSTM generative model, we create a length-$L$ RNN with basic LSTM cells. The input and output size for each cell is $K$. We feed an indicator vector of the last TF used as the input to each cell, except for the first cell, which recieves a randomly initialized variable vector as its input. The output of each cell is shifted and scaled to range from $-r$ to $r$, where $r$ is a hyperparameter. Applying the softmax function to the shifted and scaled output yields the stochastic policy: a sampling distribution over the $K$ TFs. In our experiments, we fix $r = 2$ to avoid overfitting.

**Training and Model Selection Procedure**   We trained the TF sequence generators jointly with the discriminator using SGD with momentum (fixed at 0.9), in an adversarial manner as described in [10]. We performed an initial search over the TF sequence length $L$ as described in Section 5, and then held it fixed at $L = 10$ for all subsequent experiments. We searched over a range of values for learning rates for the generator and discriminator, as well as for hyperparameters specific to our formulation, such as the diversity objective term coefficient $\alpha$, the diversity objective term distance metric $d$ (choosing between distance in the raw input space or in the feature space learned by the final pre-softmax layer of the discriminator), and whether or not to split the data used for the discriminator and generator training steps.

We selected final generators to use for test set evaluation by using them to augment training data for end classification models then evaluated on the validation set. In addition, we filtered some

generators out based on their loss (according to the discriminator $D_\phi^\emptyset$) as compared to that of random TF sequences.

**Diversity Objective**   For the diversity objective term, we tried both distance in the raw pixel-level input space and distance in the feature space learned by the final pre-softmax layer of the discriminator as choices for distance metric $d$. During training of the generators, we measured the average pairwise generalized Jaccard distance. For CIFAR-10, as an example, the final batches had an average distance of 0.52 compared to 0.86 for randomly generated sequences, which implied diversity in the learned sequences. We also computed the ratio of unique TF n-grams to total possible n-grams, and measured 0.37 compared to 0.98 for random sequences as expected.

### D.5   End Classification Models

**MNIST and DDSM**   For MNIST and DDSM we use a similar architecture to the discriminator in the previous section, adapted for the multinomial classification setting: a four-layer all-convolution CNN with leaky ReLUs and batch norm.

**CIFAR-10**   Given the flexibility of end classifier choice with our approach, for CIFAR-10 we used a more computationally expensive but standard model: a 56-layer ResNet as described in [15]. We used batch norm, regularization, learning rate schedule, and all other hyperparameters as reported in [15].

**ACE**   The end model used for the ACE task was a bidirectional recurrent neural network using LSTM cells with attention mechanisms. The maximum sentence length and attention window length were both 50. Word embeddings were initialized from pretrained vectors via [2], and updated during training. Hyperparameters were selected via a cursory grid search, and fixed for experiments.

### D.6   End Model Training

**Basic Training Procedure with Data Augmentation**   We trained all end models using minibatch stochastic gradient descent with momentum (fixed at 0.9), using a fixed learning rate schedule set once for each model and then fixed for all experiments. To perform data augmentation, during end classifier training we transformed some portion of each minibatch, $p_{transform}$. For all experiments we used $p_{transform} = 1.0$. Additionally, for the last ten epochs of training, we switched to $p_{transform} = 0.0$ following reported practice in the literature [12]. For all other hyperparameters we used default values as reported in the respective literature, held fixed at these values for all experiments.

**Transformation Regularization Term**   We additionally apply a *transformation regularization (TR)* term to the transformed data points for all image experiments by adding a term to the loss function which is the distance between the pre-softmax layer logits for each data point and its transformed copy, similar to the term in [26]. Given the fact that we are producing these transformed data points anyway, incorporating this term introduces little additional overhead.

| Task | % | Augmentation Model | TR Term Coefficient | Accuracy (Dev.) |
|------|---|--------------------|---------------------|-----------------|
| CIFAR-10 | 10 | Heuristic | 0 | 77.4 |
| | | Heuristic | 0.1 | 77.5 |
| | | LSTM | 0 | 80.4 |
| | | LSTM | 0.1 | 81.6 |

Table 4: A simple study of the effect of adding a transformation regularization (TR) term to the objective function, evaluated on a labeled validation set. We see that adding the term improves performance for both heuristic (random) TF sequences and for TF sequences generated by the trained LSTM model, and that there is a larger positive effect for the latter.

In an early calibration experiment (Table 4), we found that introducing this regularization term (using a coefficient of 0.1 and unlabeled data batch size of 20% that of the labeled data batch size) yielded improvements in performance to the end model with both learned transformation sequences and random sequences. However, we see that the positive effect is much larger for the trained LSTM

sequences (1.2 points versus 0.1 points in accuracy). We chose to subsequently keep this term fixed, viewing further calibration and exploration of this term as largely orthogonal to our central experimental questions. However, we believe that this is an extremely interesting and empirically proimising area for future study, especially given the indication that this term may be more effective when used in conjunction with a trained augmentation model such as ours.