[Reviews · NeurIPS 2017]

Reviewer 1



This paper tackles the data augmentation problem of generating more labeled training data to help the end discriminative model. It uses user-specified transformations and GAN for generating the sequence of such transformations by letting discriminator try to predict null-class. In the experimental results, the proposed approach outperforms other data argumentation baselines and the one without any data augmentation. Data argumentation is an important problem and the idea to incorporate domain-knowledge with selecting transformations by GAN is interesting. The experimental results look to improve baselines and help the cases of small amount of labeled data. However, I concern that other GAN-based approaches [26,29] also achieved comparable performance. Also I was wondering if the proposed approach could also improve performance for 100% of data sets (MNIST and CIFAR-10), or even more interesting data sets like Imagenet. Detailed comments: - Can you incorporate the end discriminative model in training? - You compare with other GAN approaches in only CIFAR-10, and how about other data sets with them? - Missing reference: Sixt et al 2016 also tries to use GAN to generate more labeled training data even though being slightly different. Typo: line #145: as as -> as Reference [1] Sixt et al, RenderGAN: Generating Realistic Labeled Data, arxiv 2016

Reviewer 2



This paper addresses an interesting and new problem to augment training data in a learnable and principled manner. Modern machine learning systems are known for their ‘hunger for data’ and until now state-of-the-art approaches have relied mainly on heuristics to augment labeled training data. This paper tries to reduce the tedious task of finding a good combination of data augmentation strategies with best parameters by learning a sequence of best data augmentation strategies in a generative adversarial framework while working with unsupervised data. Plus points: 1. The motivation behind the problem is to reduce human labor without compromising the final discriminative classification performance. The problem formulation is pretty clear from the text. 2. The solution strategy is intuitive and based on realistic assumptions. The weak assumption of class invariance under data transformations enable the use of unsupervised training data and a generative adversarial network to learn the optimum composition of transformation functions from a set of predefined data transformations. To address the challenge of transformation functions being possibly non-differentiable and non-deterministic, the authors propose a reinforcement learning based solution to the problem. 3. The experiments are well designed, well executed and the results are well analyzed. The two variants (mean-field and LSTM) of the proposed method almost always beat the basic and heuristics based baselines. The proposed method shows its merit on 3 different domains with very different application scenarios. The method experimentally proves its adaptability and usefulness of using domain specific transformation functions in the experiment with Mammography Tumor Classification experiment. The authors did a good job by showing the robustness of the method in the TF Misspecification experimentation (L279-286). Minus points: 1. I have confusion regarding the role of the discriminator which I’m writing in detail below. I can understand that the generator's task is to produce less number of out-of-distribution null class (eqn. between L113 and 114). With the weak assumption (eq.(2)), this means that the generator's task is to produce mostly an image of the same class as that of the input. Then if I'm understanding correctly, the discriminator's task is to have low value of D(.) when a transformed example is passed through the discriminator (first term of eqn between L117-L118). At the same time, the discriminator's task is to have high value of D(.) when a non-transformed example is passed through the discriminator (second term of eqn between L117-L118). Thus the 'game' being played between the generator and the discriminator is that the generator is trying to (to propose a sequence of transformations to) generate in-class (i.e., non null-class) examples while the discriminator is trying to tell whether the input coming to it is a null-class (coming from a bad generator which produces null-class example) or not (coming from the real distribution of examples or from a good generator). If my understanding (above) is true, I'd request to put some explanatory texts along this line (the game analogy and what is the role of the discriminator i.e., when will the discriminator give a high value and when low value). And if my understanding is not in the right direction, then I'd like to see a better explanation of what the authors tried to achieve by the generative adversarial objective. Side comment: Please provide eqn. numbers to all equations even if it is not referred anywhere in the text. It helps the readers to refer to them. In light of the clear motivation of a new problem, succinct solution strategy and well designed and executed experiments I’m positive about the work. However, I’d like to see the authors’ rebuttal about the confusion I wrote above. ======= After reading the fellow reviewers' comments and the the authors' response in detail, I'm more positive about this work. The authors' did a good job in clarifying my question about the role of the discriminator and have told that they would add a clarification along those lines. I agree with the authors that the focus of the work was on conducting a study of getting optimum sequence of transformation functions for data augmentation and data augmentation makes more sense when the availability of data is limited. Notwithstanding, the authors did a good job by providing the performance of their method fin the fully supervised scenario for the sake of completeness and on the request of the fellow reviewers. This also adds to the robustness of the proposed method. Thus, considering the novel problem definition, neat solution strategy and supportive experiments I am going for an even stronger accept. To me, it is a candidate for an oral presentation.

Reviewer 3



The paper proposes a technique for composing a set of given transformations to do data augmentation for supervised learning. It models the problem in a sequence modeling framework using LSTM which, given the current state, outputs at each step a probability distribution over the given set of transformations. Current state is comprised of the transformations applied so far and the transformed versions so far. The sequence model is trained using adversarial training: a "discriminator" models the distance between the distribution over transformed examples and the distribution over original examples and the sequence model is trained to minimize that distance. To avoid identity transformation, it has a diversity objective that penalizes the distance between the original example and its transformed version. To consider non-differentiable transformations, the paper uses a reinforcement learning approach (score-function based gradient estimate) to optimize the composition model. This seems to be the first work (afaik) to tackle this problem of automatically learning the order of composition the given domain specific transformations. The approach proposed is also natural and makes sense. So there is novelty in this aspect. However I am not sure if the significance of this problem is motivated well enough. The paper cites a few papers ([8,4,12]) for state-of-the-art results needing data augmentation through compositions of several transformations, without giving details on the datasets on which these SOTA results were obtained and library of transformations used there. A few specifics will be useful here (e.g on dataset X, SOTA results were obtained with model Y (resnet, etc) using random compositions of transformations A,B,C..). I also have reservations about the empirical evaluation. For most benchmark public datasets, it only considers semi-supervised learning setting (~ 10% of labeled data used for training). It will be useful to show results for fully supervised learning (with CIFAR-10 at least) to see if learning the composition for data augmentation helps at all in pushing the SOTA further. For semi-supervised learning, the results in Table 1 are not competitive I think -- MNIST with 10% labels (~500) is at 99.1 whereas SS-GAN [26] obtains about the same accuracy with 100 labels. Similarly for CIFAR-10 (table 2), the proposed approach using LSTM-RESNET obtains 81.32 with 10% labeled data (~5000) whereas SS-GAN obtains 81.37 with 4000 samples (table 2 should be corrected to reflect this discrepancy). Another difference is the classifier used -- SS-GAN uses a conv-net (though carefully designed I would guess) vs a Resnet used for the proposed method -- and some lift in the accuracy could be coming from Resnet. I also think the paper doesn't exploit the full potential of the approach for semi-supervised learning. Once a composition model is learned, it is possible to employ a smoothness regularizer using the unlabeled data for training the final classifier -- something like || f(x) - f(T(x))|| where f is the classifier and T(x) is the transformed example. This has the potential for further improving the results in Table 2 -- as it has been shown in some recent works ([1,2,3]). I like the experiments with transformation misspecification showing the robustness of the approach. Finally some more light on the use of score-function gradient estimator would be useful since it is known to have higher variance in the estimates and can finally converge to a bad solution if the model is not initialized properly. Do the authors use a more careful initialization of the LSTM model (pre-training etc)? How much time it takes to converge? What is the variance in the optimal transformation sequences (under the learned policy) for different examples? [1] TEMPORAL ENSEMBLING FOR SEMI-SUPERVISED LEARNING, 2017 [2] WEIGHT-AVERAGED CONSISTENCY TARGETS IMPROVE SEMI-SUPERVISED DEEP LEARNING RESULTS, 2017 [3] Improved Semi-supervised Learning with GANs using Manifold Invariances, 2017